# Investigation of the Fracture Process of Explosively Welded AA2519–AA1050–Ti6Al4V Layered Material

**DOI:** 10.3390/ma13102226

**Published:** 2020-05-13

**Authors:** Dariusz Boroński, Ihor Dzioba, Maciej Kotyk, Aleksandra Krampikowska, Robert Pała

**Affiliations:** 1Faculty of Mechanical Engineering, UTP University of Science and Technology, 85-796 Bydgoszcz, Poland; maciej.kotyk@utp.edu.pl; 2Faculty of Mechatronics and Mechanical Engineering, Kielce University of Technology, 25-314 Kielce, Poland; pkmid@tu.kielce.pl (I.D.); rpala@tu.kielce.pl (R.P.); 3Faculty of Civil Engineering and Architecture, Kielce University of Technology, 25-314 Kielce, Poland; akramp@tu.kielce.pl

**Keywords:** layered material, fracture process, experimental analysis, delamination, reduced temperatures

## Abstract

The study presents an analysis of the cracking process of explosive welded layered material AA2519–AA1050–Ti6Al4V (Al–Ti laminate) at ambient (293 K) and reduced (223 and 77 K) temperatures. Fracture toughness tests were conducted for specimens made of base materials and Al–Ti laminate. As a result of loading, delamination cracking occurred in the bonding layer of specimens made from Al–Ti laminate. To define the crack mechanisms that occur at the tested temperatures, a fracture analysis was made using a scanning electron microscope. Moreover, acoustic emission (AE) signals were recorded while loading. AE signals were segregated to link their groups with respective cracking process mechanisms. Numerical models of the tested specimens were developed, taking into account the complexity of the laminate structure and the ambiguity of the cracking process. A load simulation using the finite element method FEM allowed calculating stress distributions in the local area in the crack tip of the Al–Ti laminate specimens, which enabled the explanation of significant material cracking process development aspects. Results analysis showed an influence of interlayer delamination crack growth on the process of the Al–Ti laminate specimen cracking and the level of K_Q_ characteristics at different temperatures.

## 1. Introduction

Explosive welding makes it possible to connect materials with different mechanical and metallurgical properties into one layered material. The explosive welding of base materials requires a very large amount of kinetic energy to be applied to the layer through the controlled explosion of an explosive (high energy material) [1].

Layered composites, due to highly diverse properties of the component materials used to form the layers, allow achieving materials whose functional qualities are different from those of the base materials. This applies not only to mechanical properties, but also to other properties such as corrosion resistance [2], thermal conductivity [3], or electrical conductivity [4,5]. A literature analysis indicates that there is a significant number of different metals used in explosive welded layered composites, which enables producing different configurations, such as Al/Al [6], Al/Cu [7,8], Al/Mg [9,10], Al/Fe [11,12], Al/steel [13,14], Al/Ni [15], Ti/Mg [16], Ti/Ni [17,18], Ti/Cu [19], and Ti/steel [20,21].

Explosive welding was used in constructing a new structural layered material, AA2519–AA1050–Ti6Al4V, which was developed in cooperation with the Explomet company, scientific units of the Military University of Technology in Warsaw, Warsaw University of Technology, Institute of Non-Ferrous Metals, Space Research Centre of the Polish Academy of Sciences, and UTP University of Science and Technology in Bydgoszcz. It is designed for use in the aviation and aerospace industries and technical objects exposed to ballistic loading.

Fracture toughness is a crucial property of construction materials, particularly those which are used in objects that are susceptible to fractures when manufactured or used. The fracture toughness of homogeneous materials can be considered in terms of the material properties (fracture toughness characteristics), taking into account all the limitations connected with the specimen shape and dimensions [22,23]. This is, however, much more complicated for non-homogeneous materials, including layered composites.

The mechanical properties of layered materials, including fracture toughness, depend on the nature of the bonds formed in the connection zone. Different approaches to model the interface zones of layered materials can be found in the literature. A comprehensive and detailed review of modelling and characterization of interface properties was discussed in the paper [24]. Authors presented nano and micro-mechanical approaches to modelling the interfacial bonding strength. They stated that in the case of properties such as bonding strength, a small number of models exist with limited predictability. In the absence of a reliable bond model experimental analysis should be applied to achieve an effective bonding strength of the interface. This also applies to explosively welded layered materials, which interface properties depend on joined materials and process parameters [1,25,26].

The literature provides different approaches to the fracture toughness analysis of layered composites, where the focus is put on assessing the resistance to the growth of a crack that occurs between the layers (interfacial crack) determined using tensile bending, or impact tests.

For example, [27] includes results of fracture toughness tests for compact tension (CT) specimens made of hot roll bonded clad pipes based on two 316-L austenitic stainless steel-X60 (base metal) / X65 (clad) carbon steel with and without an Ni-interlayer. The interlayer was in the crack path of the CT specimen. Tests were performed in air and under in situ electrochemical hydrogen charging. The highest fracture toughness was obtained for the samples without a Ni-interlayer tested in air, followed by the samples with an Ni-interlayer tested in air.

Fracture toughness tests on the metal-clad interface of API X52 pipes cladded with an Inconel 625 alloy by welding overlay are presented in work [28]. CT specimens with the notch tips machined as close as possible to the interface plane were used in the tests. The findings indicate that the interface region has high toughness and the crack tip opening displacement resistance curves shapes (CTOD-R) are characteristic of structural metallic materials. Reference [29] provides an analysis of the impact of interface oxides on the shear properties of hot-rolled clad plate made of HSLA steel and 316 L stainless steel. The results were that high levels of interface oxidation tended toward low fracture toughness failure during the process of deformation.

Reference [30] presents fracture toughness tests for a welded layered material, St37 Steel-Ck60, using an impact test. It was found that fracture toughness of a layered material was higher than that of a flyer material due to the higher fracture toughness of the base material.

Not always, however, the direction of loading to the surface of elements made of layered materials is perpendicular. Reference [31] presents fracture toughness tests for a layered Al/Al material made of ultra-fine grained aluminum produced through an accumulative roll bonding (ARB) process, whose loading was carried out within the layer connection area. By increasing the number of the ARB cycles, the value of fracture toughness increased up to 155% compared to an annealed sample.

In work [32], one can find a comprehensive analysis of a testing methodology and the influence of different factors on the fracture toughness of layered materials made of aluminum-lithium alloys used in the aerospace industry. One of the conclusions to be formulated on the basis of this work is that fracture toughness is a critical property that needs to be considered when selecting materials for use in aerospace applications which results not only from the need to reduce the risk of the unstable cracking of layered structures in use but it is also necessary to provide them with “crashworthiness”.

The literature does not provide an unequivocal assessment of the fracture toughness of explosive welded aluminum and titanium alloys. Some studies prove that, in general, explosive welding allows a good fracture toughness due to the particular structural state of the welds formed by the dynamic interaction of workpieces [33,34]. However, at the same time, brittle intermetallics, characterized by low fracture toughness, can occur in the weld zone [35].

Considering the above, tests of AA2519–AA1050–Ti6Al4V layered material were carried out in laboratory conditions at 293 K and at temperature reduced to 77 K. The fracture toughness was tested for the analyzed layered material compared to the base materials to find the influence of temperature. Taking into consideration the complexity of its structure, numerically modeling the specimens was used to analyze the cracking process, and numerical loading simulations were carried out. This, in turn, allowed defining stress and strain distributions in the front of the specimen under loading. Acoustic emission (AE) signals were recorded during loading. Applying signal grouping methods allowed distinguishing a few major groups of AE signals that can be associated with certain characteristic mechanisms that occur in the layered material cracking process. The cracking process was also studied using scanning electron microscopy for fracture surface analysis. Using different investigation methods enabled a comprehensive analysis of the cracking process in AA1519-AA1050-Ti6Al4V layered material, including its description.

## 2. Materials and Methods

The main goal of this study is an analysis of the cracking process of AA2519–AA1050–Ti6Al4V (Al–Ti laminate) layered material. The base materials used for constructing the Al–Ti laminate were AA2519 aluminum alloy and Ti6Al4V titanium alloy. AA2519 aluminum alloy is a relatively new structural material with the chemical composition given in Table 1 [36]. The mechanical properties of the alloy at ambient temperature (293 K) and cryogenic conditions (77 K) are given in Table 2.

Excellent mechanical properties, including high impact strength and ballistic resistance, are obtained by precipitation hardening the alloy. Owing to such properties, it is applied in constructing ballistic protection shields for light military vehicles because of the possibility of reducing the weight and, thus, improving their mobility.

AA2519 aluminum alloy was subjected to pre-treatment involving hot rolling and annealing at 400 °C for 1 h to increase plasticity and reduce internal stress, which makes the alloy easier to weld. In this way, a coarse-grained structure with large homogeneously distributed Al_2_Cu (phase θ) particles [25] was obtained.

The second material used in the analyzed layered composite was Ti6Al4V titanium alloy. Its chemical composition is given in Table 3, and its mechanical properties determined at ambient temperature (293 K) and cryogenic conditions (77 K) based on static tensile diagrams are presented in Table 4. Owing to high strength at relatively good plasticity, this alloy finds a wide application as a structural material, among other uses in aviation, especially for machined load-carrying components of an airframe structure and drive systems. Ti6Al4V alloy has an α + β type structure, which consists of coarse grains of α phase and β phase rich in vanadium and aluminum precipitations located at the borders of the grains [25].

Another material used in constructing the Al–Ti laminate was AA1050 aluminum alloy, the chemical composition of which is given in Table 5. The thin layer of AA1050 alloy was a technological spacer (interlayer) designed to reduce the potential brittleness of the intermediate Al–Ti zone created by the explosive welding.

The tested layered material was 10 mm thick. Explosive welding of Ti6Al4V titanium alloy and AA2519 aluminum alloy was carried out by the Explomet Company. Details on the welding technology were presented in [37,38]. The explosive Saletrol (based on ammonium sulfate and hydrocarbon fuel) was used in the welding process. Testing plates were produced using the explosive material at a detonation velocity in the range of 1850–2000 m/s, at variable bonding parameters falling within the range of 420–620 m/s (for plates collision speed) and a collision angle of approximately 15°.

The parallel plating configuration was applied, where the base layer was a 5 mm thick Ti6Al4V alloy sheet and the overlaid layer (flayer) was a 5 mm thick AA2519 alloy sheet with an approximately 0.2 mm thick unilaterally rolled soft layer of AA1050 aluminum alloy. The distance between the welded layers was 5 mm. As a result of the explosive welding, a complex structure was created on the Al–Ti border, the construction of which is described in detail in [25]. 

The fracture toughness tests of base materials and plater in ambient and cryogenic conditions were performed using CT type specimens. The testing methodology used in cryogenic conditions and the initial results of fracture toughness verification tests for Al–Ti laminate were presented in conference proceedings [39].

Tests at a lowered temperature were performed using a special environmental chamber. It enabled continuous specimen cooling through immersion in liquid nitrogen. In effect, the approximate specimen temperature was 77 K. Images of the specimens taken during the tests and their dimensions are shown in Figure 1. The specimens were prepared using the wire electrical discharge machining method.

The tests were conducted using an Instron hydraulic testing machine, with a 0.02 mm/s controlled displacement value. The crack-opening displacement extensometer base was 10 mm, while its working range was ± 4 mm.

Tests were also performed using prismatic specimens with a one-sided notch (crack) loaded according to the three-point bend test scheme (Single Edge Notched Bend (SENB) specimen) with dimensions B = 10 mm, W = 20 mm, S = 80 mm, and a/W ≈ 0.5 (Figure 2a).

The specimens were tested at ambient temperature (293 K) and a reduced temperature (223 K). The tests were supposed to compare the results of the cracking process for two different specimen types, CT and SENB, and record AE signals during the specimen loading process (Figure 2c). A MISTRAS apparatus was used for recording AE signals [40]. All procedures applied in specimen preparation, test performance, and signal recording, as their analyses, were performed according to ASTM [41] standards. The tests were conducted on a Zwick-100 strength testing machine using a thermal chamber (Figure 2b).

Nitrogen vapor was used as a coolant. The lower temperature was set at 223 K due to the limited range of AE sensor applicability. Fracture surfaces of the tested specimens were observed using a JEOL scanning electron microscope.

## 3. Results and Discussion

### 3.1. Fracture Toughness

To define the fracture toughness of specimens made of base materials and a specimen made of the Al–Ti laminate in ambient and cryogenic temperature, tests were performed to determine the critical value of stress intensity factor (SIF) in the linear–elastic range. Tests were conducted for specimens CT (Figure 1a) and SENB (Figure 2a) with a fatigue generated initial crack length ranging from 0.45–0.55 W.

It needs to be mentioned that the tests were conducted to compare fracture toughness parameters, rather than determine their normative values, which could be treated as the material characteristics in the structure analysis methods. It applies mainly to a layered material for which the process of cracking may take a completely different course with layers interacting.

The force value changes were determined from the test results as a function of crack opening, as presented in Figure 3 and Figure 4.

The values of P_Q_, P _max_, P _max_/P_Q_, and K_Q_, determined from the tests are presented in Table 6.

An analysis of the diagrams shown in Figure 3 and Figure 4 and the data from Table 6 indicates that the specimens cracked in different ways. In the case of AA2519 alloy specimens, a distinct plastic character can be noticed for the cracking process regardless of the testing temperature. The determined critical values, characterizing fracture toughness K_Q_, have similar levels for both tested temperatures, which would indicate no temperature influence on the process of AA2519 alloy cracking within the tested temperature range, from 293 to 77 K.

Cracking more similar to brittle cracking was observed for Ti6Al4V specimens, however, mainly at cryogenic conditions. Temperature reduction also results in reducing the value of P_Q_, force, and K_Q_.

Comparing the value of K_Q_ for AA2519 and Ti6Al4V materials and AA2519–AA1050–Ti6Al4V layered material indicates a distinctly higher toughness for the titanium alloy than the aluminum alloy, both in ambient conditions and cryogenic ones. The fracture toughness of the layered material expressed by K_Q_ is similar to the fracture toughness of AA2519 alloy for both test temperatures.

The results of tests on SENB specimens made of the Al–Ti laminate in ambient temperature are similar to those obtained for CT specimens (Table 6). Since the stiffness of CT and SENB specimens is different, the loading line slope is also different. Additionally, the character of the P–COD diagram is similar. The P_max_/P_Q_ ratio exceeds the permissible normative level of P_max_/P_Q_ < 1.1 for all SENB specimens, whereas it approaches it for CT specimens. These diagrams can indicate a high level of material plasticity or non-homogeneity for the process of cracking, showing the occurrence of different cracking mechanisms.

Another aspect that needs attention is the inappropriate stiffness change in the Al–Ti laminate specimen. Temperature reduction in the elements made of these materials usually causes an increase in their stiffness, which is the effect of an increase in the values of strength characteristics, primarily Young’s Modulus [42,43,44]. The diagrams of CT specimens made of a single monolithic material (Figure 3a,b) show that this tendency proves to be less true for Al alloy and truer for Ti alloy. However, in the case of CT specimens made of laminate, an increase in stiffness was observed for only one of them at cryogenic temperature (which could have been caused by a shorter crack), whereas no increase was found for the two remaining ones (Figure 3c). A decrease in stiffness was observed for SENB specimens tested at reduced temperatures compared to those tested at ambient conditions (Figure 4), along with the earlier mentioned loading line deviation from the straight-line trend. Inconsistencies between P–COD diagrams and the expected ones, observed during the specimen loading, gave reasons to carry out further research to explain the cracking mechanisms of layered material.

### 3.2. Numerical Modeling, Loading Simulation and Calculating Stress Fields

To analyze the stress state in the area in front of the initial crack tip, numerical modeling was performed using the finite element analysis program ABAQUS for an Al–Ti laminate SENB specimen.

Numerical calculations need an appropriate constitutive equation definition between true values of strains and stresses, σ_t_ = f(ε_t_) because it determines the correctness of the results. Determining the σ_t_ = f(ε_t_) dependence is based on the calibration procedure carried out based on experimental tests with different configurations of specimens made of the same material to provide different levels of triaxiality stress factor and Lode coefficient [45,46,47,48]. The procedure can be used for homogenous materials, however, it cannot be used for layered composites. For the Al–Ti laminate SENB specimen calculations, the σ_t_ = f(ε_t_) dependence was created based on the respective dependencies of base materials. Nominal dependencies σ_n_ = f(ε_n_) were obtained from the results of uniaxial tensile tests of base materials [31], which provided the basis for calculating true values on the uniform extension section (up to the beginning of necking), according to dependences ε_t_ = ln(1 + ε_n_) and σ_t_ = σ_n_(1 + σ_n_). Dependence σ_t_ = f(ε_t_) was extended to higher strain values using a straight line dependence obtained from approximating the dependence’s last 250–300 points. Dependencies σ_t_ = f(ε_t_) were approximated to the value of strains ε_t_ = 3.0. So, constitutive equations defined for respective base materials were introduced into the numerical model.

Due to the symmetry plane, half of a three-dimensional SENB specimen was modeled (Figure 5a). The specimen symmetry conditions were assumed to be in the Y-direction in relation to the XZ plane, which overlaps with the cracking plane. A support roller was locked, which enabled displacing the specimen in the X-direction. The specimen was divided into 22 layers in the direction of the thickness. The tip of the crack was represented in the form of a quarter of a circle with a 12 μm radius, which was divided into 12 equal parts (Figure 5b). The dimensions of the finite element mesh decreased in the radial direction up to the tip of the crack.

The modeled half of the SENB specimen was divided in the direction of thickness into two equal parts that were assigned real stress–strain curves. A model of large strains was assumed for calculations.

Loading was defined using loading roll displacement. The value of the loading roll displacement was recorded during experimental tests using an extensometer.

Distributions of the stress components at 293 K and 223 K are shown in Figure 6a,b, respectively. The calculations were performed for the moment of loading, which is consistent with the occurrence of P_max_ force in the specimen. Maximal levels of stress occur at about 0.136 mm from the crack tip, and they decrease further. At the reduced temperature (223 K), the level of stress components is higher by 200–300 MPa than at ambient conditions (293 K). 

The obtained distributions of stress components show significant diversity for the respective layers of Al–Ti laminate. In the Ti6Al4V layer, the level of stress components is higher by 800–1400 MPa than in the AA2519 layer. Whereas, along with the temperature decrease, the gradient (difference) of stresses between the base layers increases. For the reduced temperature, the difference in the stress levels is higher by about 200 MPa than for ambient temperature. So high stresses can cause the growth of an interlayer crack and formation of internal delamination cracks.

### 3.3. Specimen Fractography

#### 3.3.1. Observation of Delamination Cracking

Observing the fractures of tested Al–Ti laminate specimens confirms the above supposition that internal interlayer cracks can occur during loading (Figure 7). Delamination cracks were found in the specimens tested at ambient (293 K), lowered (223 K), and cryogenic (77 K) temperatures. 

The presence of interlayer delamination cracks encouraged a deeper exploration of the explosive welded area. For this purpose, tests of the specimen microstructure and fracture surfaces were performed using a scanning electron microscope. Figure 8a shows a weld that consists of three layers. The bottom part is AA2519 aluminum alloy, the upper part is Ti6Al4V titanium alloy, and the middle is AA1050 aluminum alloy. As has already been mentioned, the 200 µm thick middle part of AA1050 alloy, was a technological spacer (interlayer) designed to reduce the potential brittleness of the intermediate Al–Ti zone created by the explosive welding. The weld between the AA1050 and AA2519 aluminum alloy layers was made by the plastic machining (rolling) method, is continuous, and does not exhibit visible defects. The nature of the connection between the Ti6Al4V and AA1050 layers in the explosive welding zone is slightly different. Line scan EDS analysis (Figure 8c,d) showed that the 20–40 µm wide transition zone (TZ) between Ti6Al4V and AA1050 is rich in aluminum and titanium, which means that both materials mix strongly during welding. Moreover, in the transition zones, numerous isles occur, which contain mainly Ti6Al4V alloy and Al_3_Ti or Ti_3_Al intermetallic phases formed by the explosive welding process [25]. Additionally, single voids and discontinuities can be observed in the material (Figure 8b).

Observation of fractures in a specimen of CT and SENB layered material allowed one to find the occurrence of delamination on both sides of AA1050 interlayer. Transitions of delamination cracks from the AA2519 layer to the Ti6Al4V layer, and the other way round, could be observed. Simultaneous delamination of the specimen on both sides of AA1050 was rarer.

However, the growth of delamination cracks between the Ti6Al4V and AA1050 layers was prevalent, particularly in SENB specimens. This is caused by the occurrence of different types of precipitation particles containing intermetallic compounds and the presence of material discontinuities in the form of voids and microcracks. The Ti6Al4V layer is also characterized by a high level of stress (Figure 6), which causes cracking of precipitation particles and subsequent crack growth. Observations of delamination cracks indicate that their initiation usually starts with precipitation particle cracking. The microcracks then grow using the ductile mechanism. After crack formation in the area situated to the closer titanium alloy layer, cracks in the AA1050 layer propagate in the direction from Ti to Al (Figure 9), using the shearing mechanism. 

#### 3.3.2. Fracture Surface

• AA2519 Aluminum Alloy

A comparison of the fracture surfaces of CT specimens made of AA2519 aluminum alloy tested in ambient and cryogenic conditions is presented in Figure 10. 

The analysis indicates two major mechanisms of AA2519 alloy cracking in ambient and cryogenic conditions-nucleation, growth, and coalescence of micro-voids and inter-crystalline fractures. Large cavities are dominant all over the fractures at ambient temperature, which is the effect of connecting voids formed by large, unsolved phase θ (Al_2_Cu) particles, characteristic for precipitation-hardened aluminum alloys. They are accompanied by medium voids associated with small particles. Between the primary cavities are void sheets derived from the particles of the dispersion phase, which increase and connect during the coalescence of primary voids. The smallest group are ductile cracks on the boundaries of grains and between grains, represented by numerous, very small cavities on facets that occur on the fracture. The distribution of all types of cavities is similar throughout the fracture length.

At cryogenic conditions, the share of large primary voids decreases in favor of inter-crystalline cracks, particularly for the crack growth initial phase, that is, near the boundary of the fatigue crack. The crack length growth is accompanied by an increase in the number of larger cavities. Additionally, transverse cracks can be observed in the θ phase on the fractures of the specimens tested in cryogenic conditions. For ambient conditions, this phenomenon did not occur.

• Ti6Al4V Titanium Alloy

Fractures of CT specimens made of Ti6Al4V titanium alloy are presented in Figure 11. They are of similar character for both temperatures, though certain differences can be noticed. In both analyzed cases, the form of the fracture is typical for a two-phase titanium Ti6Al4V. It needs to be emphasized that at both temperature conditions, flattened areas occur, which indicate quasi-brittle cracking. They are particularly visible at 77 K. In both fractures, there are also cleavage planes formed around the cavities, flattened areas and surfaces created by the above described quasi-cleavage fracture.

The fracture topographies at ambient conditions are more diversified. The character of the fractures is similar throughout the crack length for both temperatures.

• AA2519–AA1050–Ti6Al4V Laminate

Figure 12 and Figure 13 show images of fracture surfaces for AA2519 and Ti6Al4V layered material. Both for aluminum alloy and titanium alloy, the character of fracture topography obtained in the layered material is similar to those of homogeneous specimens.

However, the process of explosive welding reinforces the component materials, which is reflected in the character of the fracture surface, that is, the height and reduction in some topographic details characteristic of ductile cracking.

The fracture relief of the AA2519 layer stands out only due to a slightly smaller number of large voids compared to the number of medium and small voids, which is naturally associated with a larger number of cracks along the grain boundaries. 

The basic difference of the fracture relief found in the Ti6Al4V layer is their slightly less diversified topography for both temperatures.

### 3.4. Acoustic Emission Testing of The Layered Material Fracture Process

The tests discussed in the previous sections indicate the complexity of the Al–Ti laminate cracking process. AE signals were recorded to explain some ambiguities of the cracking process for SENB specimens made of laminate. The runs of P–COD loading diagrams for selected SENB specimens tested at ambient temperature (293 K) and reduced temperature (223 K) are presented in Figure 14 along with recorded AE signals. 

The highest power AE signals, above 6.0 × 10^7^ (pV × s), occur in both specimens right after the maximum loading and are probably caused by the mechanisms of ductile growth of subcritical cracks in the layers of AA2519 and Ti6Al4V material. 1.0–6.0 × 10^7^ (pV × s) signals occur in a slightly different way at different temperatures. For the specimens tested at ambient temperature (293 K), the great majority of signals from this interval occur after the loading maximum. Whereas, for the specimens tested at reduced temperature (223 K), more signals from this range occur before the maximal force value. Small power signals, lower than 1.0 × 10^7^ (pV × s), with amplitude level below 40 dB, can be well identified with those coming from mechanical noise.

A comparison of AE signals with the shape of P–COD loading diagrams shows the differences between the cracking process for specimens made of Al–Ti laminate at different test temperatures [49]. At ambient temperature (293 K), the growth of a delamination interlayer fracture was initiated and continued along with the development of the main subcritical fracture in the base layers. However, at reduced temperature (223 K), interlayer cracking was initiated much earlier and until the moment the maximal force was achieved; the delamination crack had already been formed.

## 4. Conclusions

The study presents the results of tests of the cracking process for Al–Ti laminate at ambient conditions (293 K) and reduced temperatures (223 and 77 K). Microscopic analysis of fractures, AE, and numerical modeling were used in the cracking process analysis. A few conclusions can be drawn based on an analysis of the test results.

The cracking process of AA2519–AA1050–Ti6Al4V layered material is more complex than that of homogenous metallic materials. In Al–Ti laminate specimens, subcritical fractures developed in the main crack plane and perpendicularly to it, causing their delamination. Since normalized methods for determining fracture toughness characteristics were developed for homogeneous material specimens, where the subcritical fracture growth develops in the same plane as the initial fracture, in the case of a layered material, the values obtained with their use can be applied only for comparative purposes. Having this in mind, the K_Q_ value determined for CT and SENB specimens of the Al–Ti laminate slightly decreased with the temperature reduction (Table 6).The character of the fracture surfaces of the Al–Ti laminate layers was similar to the cracks observed in homogeneous specimens, both at ambient and lowered temperatures. An analysis of the specimen fractures showed, however, the formation of delamination cracking in both testing conditions.Interlayer delamination crack growth has a large influence on the process of the Al–Ti laminate specimen cracking and the level of K_Q_ characteristics. The test results indicate that at reduced temperature, an interlayer fracture starts growing during specimen loading, and when the specimen is exposed to maximal loading, it has already been formed, whereas a drop in the force value is caused by the growth of a subcritical fracture in the base material layers (AA2519 and Ti6Al4V). At ambient temperature, the delamination processes and subcritical crack growth occur almost at the same time once the maximum loading is applied to the specimen. The above was proven by the test results obtained in the study:-According to the numerical calculations, stresses reach higher levels in specimens tested at a reduced temperature, hence the difference between the stress levels in Ti6Al4V and AA2519 layers is higher compared to ambient temperature;-A smaller inclination angle and deviation from linearity for the P–COD loading line for specimens tested in reduced temperatures indicates a decrease in the specimen stiffness, which is the effect of forming interlayer delamination cracks as early as at the specimen loading stage;-When the specimens are loaded at reduced temperature, numerous AE signals occur, which indicates an interlayer cracking process; and-AE signals were recorded for the specimens tested at ambient conditions, mainly for the maximal value of the force and at the stage of the crack growth, which means that growth of the delamination and subcritical cracking occurred at the same time.

## Figures and Tables

**Figure 1 materials-13-02226-f001:**
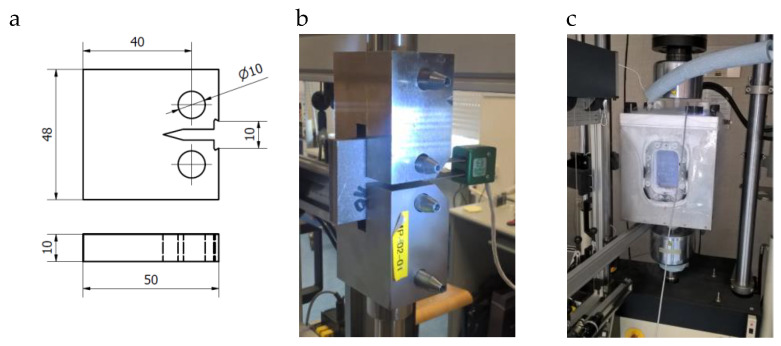
Specimen and stand: (**a**) specimen geometry, (**b**) testing in ambient temperature, and (**c**) test with specimen immersed in liquid nitrogen.

**Figure 2 materials-13-02226-f002:**
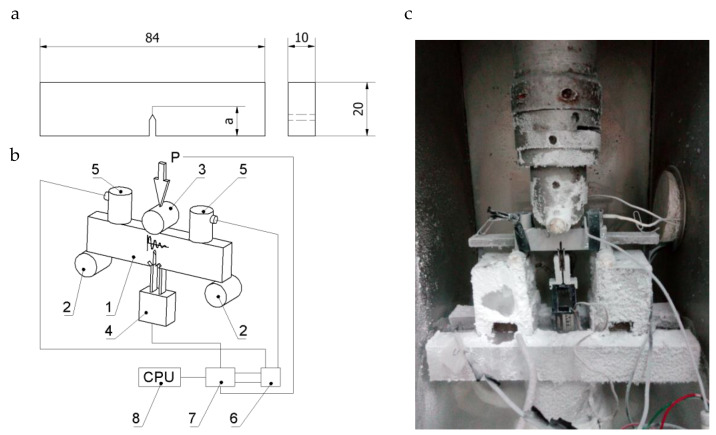
Schematic of an Single Edge Notched Bend (SENB) specimen (**a**). Schematic (**b**) and view (**c**) of the acoustic emission (AE) sensors placed during of the SENB specimen loading: 1—SENB specimen, 2—support rolls, 3—loading roll, 4—COD extensometer, 5 and 6—AE sensors, 7—transducer, 8—conditioner, and 9—PC.

**Figure 3 materials-13-02226-f003:**
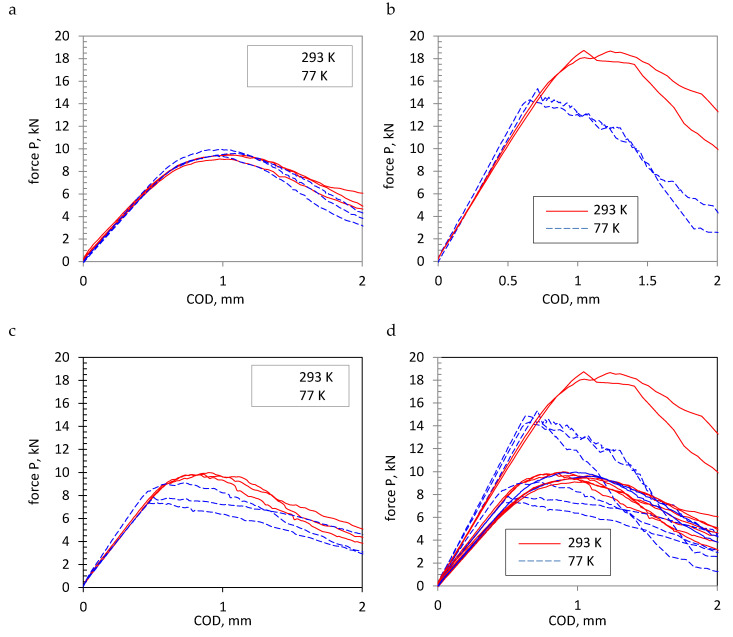
Diagrams of compact tension (CT) specimen loading in ambient and cryogenic temperatures: (**a**) AA2519 aluminum alloy, (**b**) Ti6Al4V titanium alloy, (**c**) AA2519–AA1050–Ti6Al4V layered material, and d) presentation of all test results.

**Figure 4 materials-13-02226-f004:**
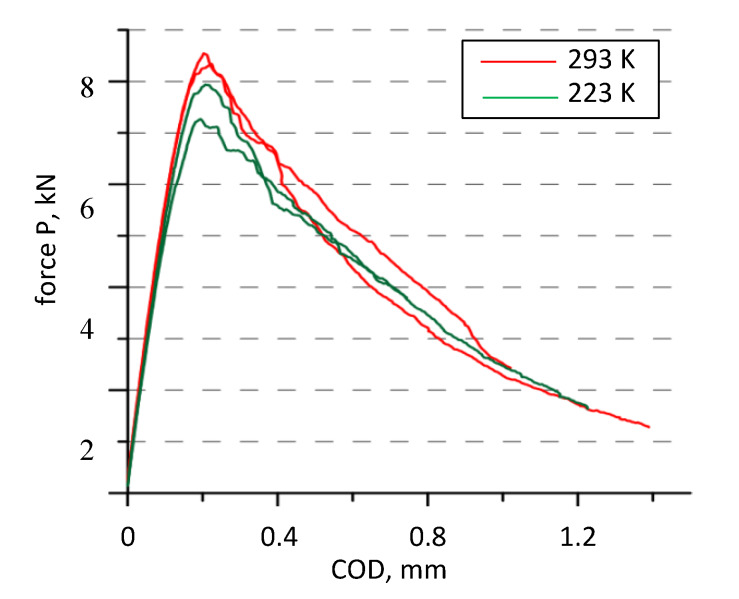
Diagrams of SENB specimen loading at ambient (293 K) and lowered (223 K) temperatures.

**Figure 5 materials-13-02226-f005:**
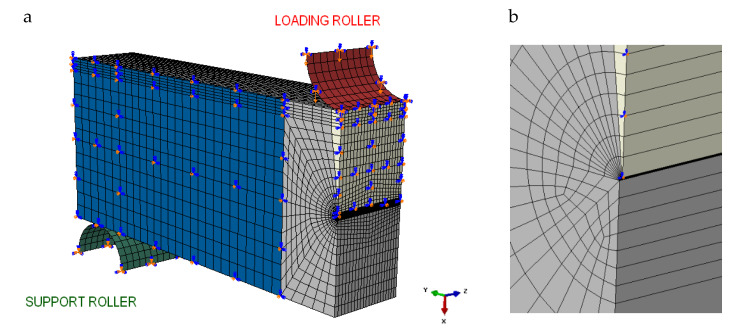
A mesh used in numerical modeling and calculations for the SENB specimen (**a**) and crack tip zone (**b**).

**Figure 6 materials-13-02226-f006:**
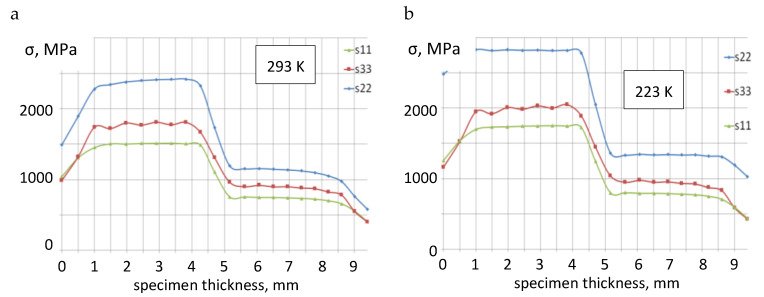
Stress distributions in front of the crack tip: (**a**) for ambient temperature, 293 K and (**b**) for reduced temperature, 223 K.

**Figure 7 materials-13-02226-f007:**
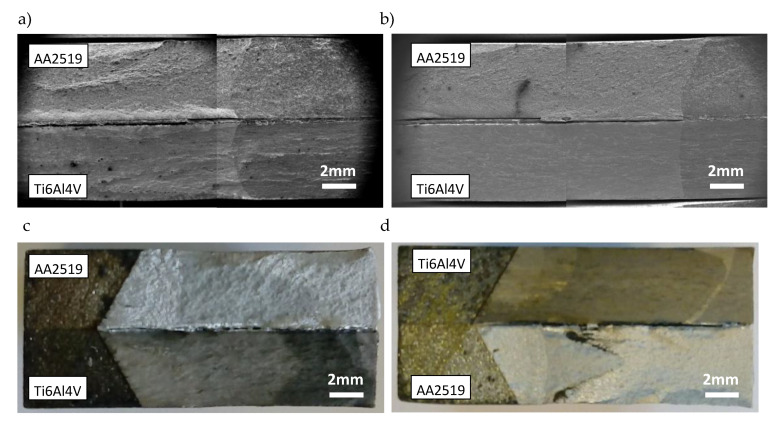
Delamination cracking in the weld area: (**a**) CT specimen-293 K, (**b**) CT specimen-77 K, (**c**) SENB specimen-293 K, and (**d**) SENB specimen-223 K.

**Figure 8 materials-13-02226-f008:**
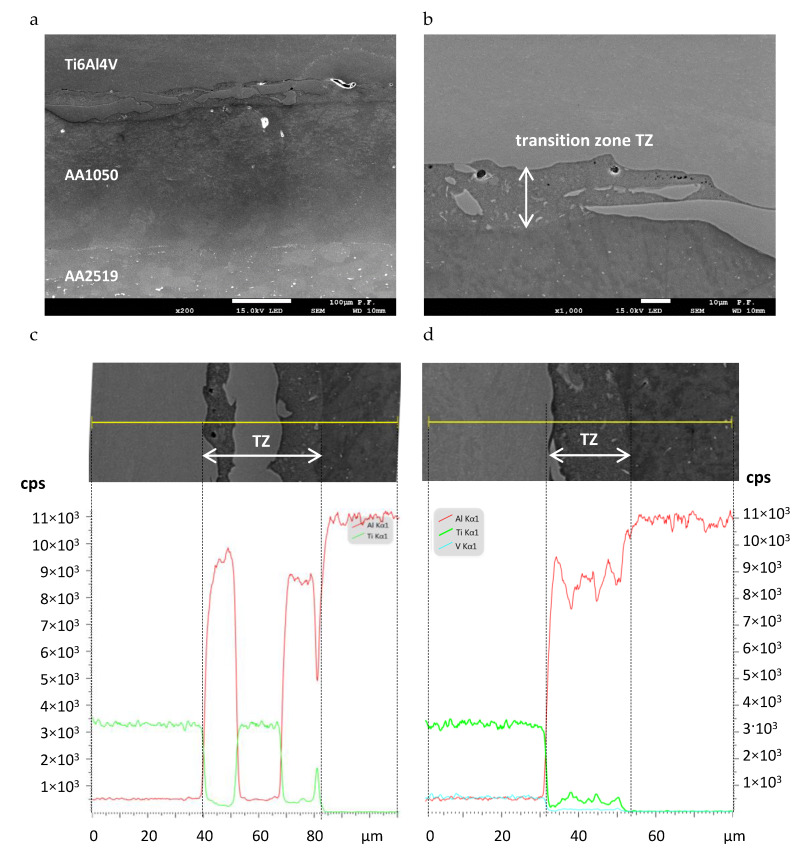
Microstructure of AA2519–AA1050–Ti6Al4V: (**a**) bonding zone, (**b**) transition zone of AA1050-Ti6Al4V and (**c**,**d**) line scan EDS analysis of transition zones.

**Figure 9 materials-13-02226-f009:**
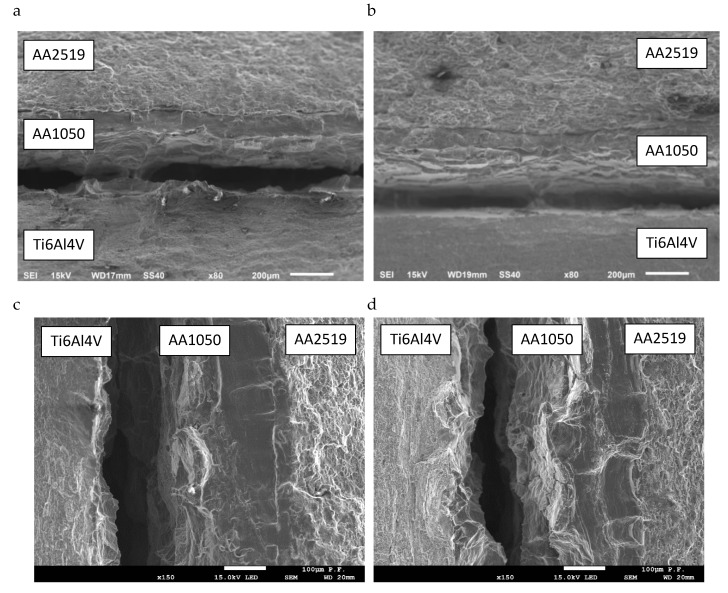
Delamination cracking in specimens tested at a different temperature: (**a**) CT-293 K, (**b**) CT-77 K, (**c**) SENB-293 K, and (**d**) SENB-223 K.

**Figure 10 materials-13-02226-f010:**
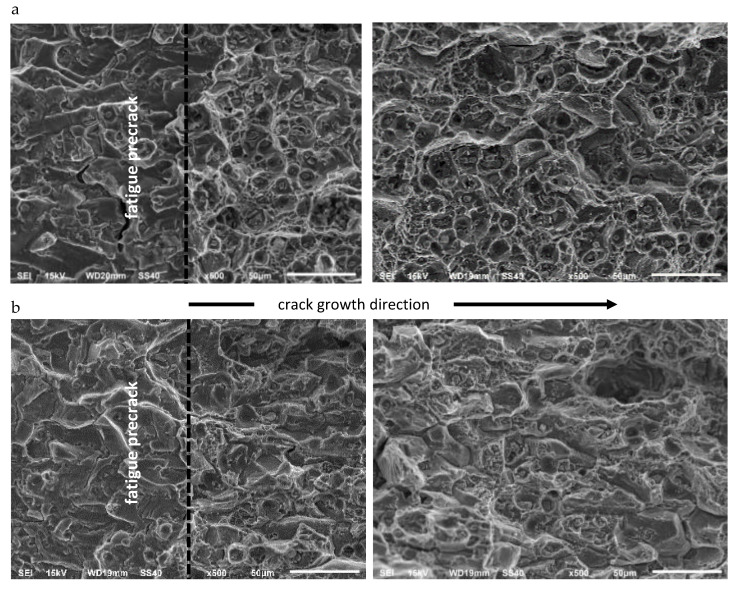
A comparison of fracture surfaces of CT specimens made of AA2519 aluminum alloy at 293 K (**a**) and at 77 K (**b**).

**Figure 11 materials-13-02226-f011:**
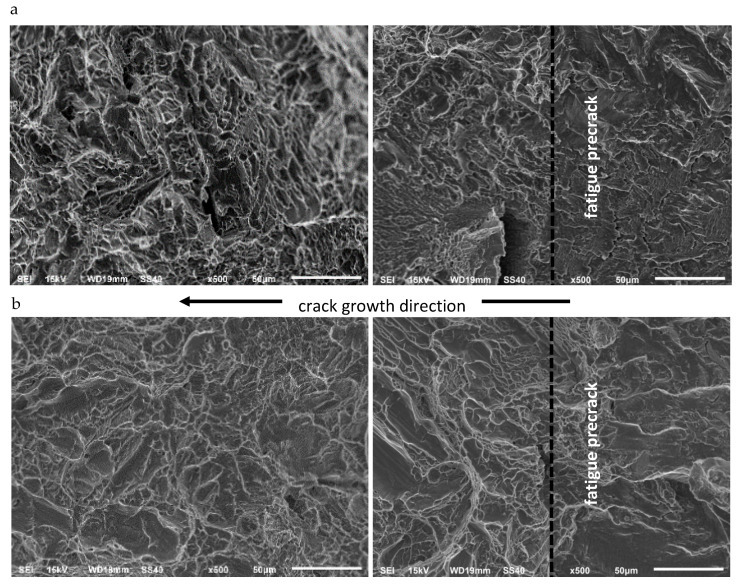
A comparison of fracture surfaces of CT specimens made of Ti6Al4V titanium alloy at 293 K (**a**) and at 77 K (**b**).

**Figure 12 materials-13-02226-f012:**
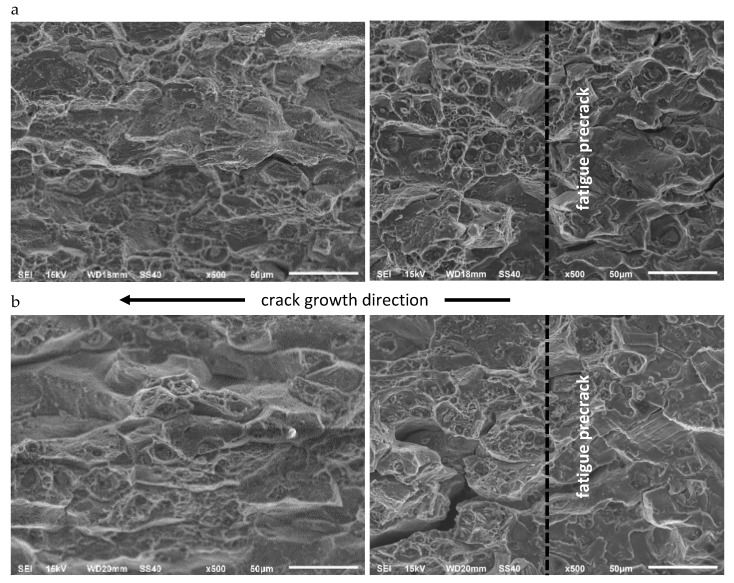
A comparison of AA2519 layer fractures relief in CT specimens made of Al–Ti laminate at 293 K (**a**) and at 77 K (**b**).

**Figure 13 materials-13-02226-f013:**
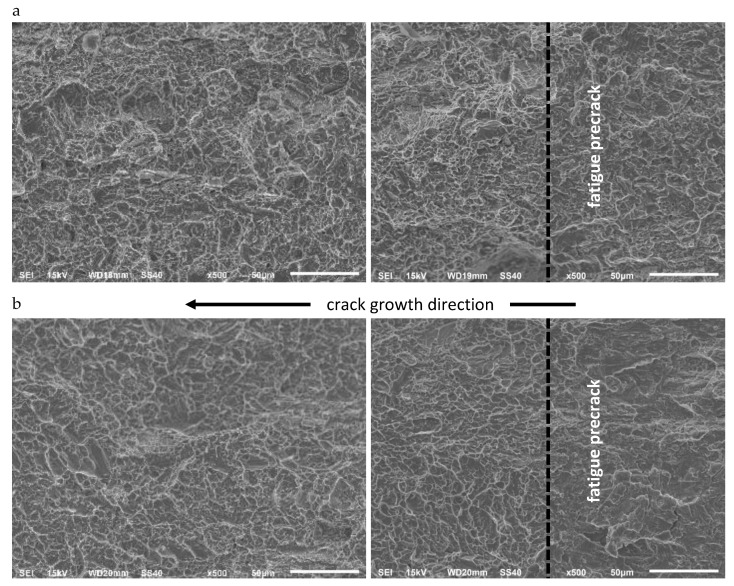
A comparison of Ti6Al4V layer in CT specimens made of Al–Ti laminate at 293 K (**a**), at 77 K (**b**).

**Figure 14 materials-13-02226-f014:**
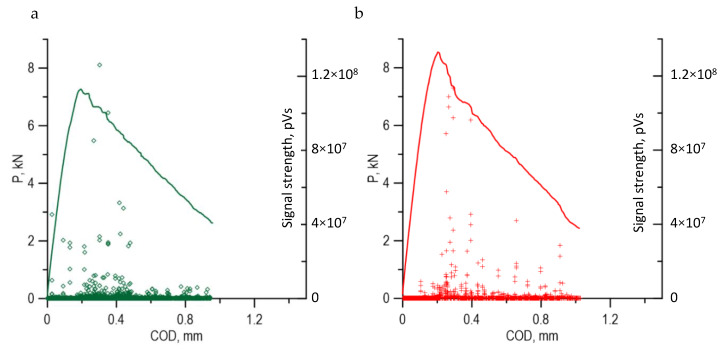
Diagrams of P–COD loading with EA signals for 293 K (**a**) and 223 K (**b**).

**Table 1 materials-13-02226-t001:** Composition of AA2519 [36].

Chemical Composition, wt. %									
Si	Fe	Cu	Mg	Zn	Ti	Sc	Zr	V	Al
0.06	0.08	5.77	0.18	0.01	0.04	0.36	0.12	0.12	balance

**Table 2 materials-13-02226-t002:** Mean values of the mechanical properties of AA2519 aluminum alloy at ambient and cryogenic condition [37].

Temp.	σ_y_	σ_u_	E	A_5_
K	MPa	MPa	GPa	%
293	353	475	67.5	16.3
77	426	595	80.3	19.1

**Table 3 materials-13-02226-t003:** Composition of Ti6Al4V [25,36].

Chemical Composition, wt. %							
O	V	Al	Fe	H	C	N	Ti
<0.2	3.5	5.5	<0.3	<0.0015	<0.08	<0.05	Balance

**Table 4 materials-13-02226-t004:** Mean values of the mechanical properties of Ti6Al4V titanium alloy at ambient and cryogenic condition [37].

Temp.	σ_y_	σ_u_	E	A_5_
K	MPa	MPa	GPa	%
293	859	908	111.7	13.6
77	1344	1392	128.6	12.1

**Table 5 materials-13-02226-t005:** Composition of AA1050 [25].

Chemical Composition, wt. %							
Si	Fe	Cu	Mg	Mn	Ti	Zn	Al
0.25	0.4	0.06	0.05	0.05	0.05	0.07	balance

**Table 6 materials-13-02226-t006:** Fracture toughness test results in the linear-elastic range.

Material	Specimen	Temp.	P_Q mean_	P_max mean_	(P_max_/P_Q_) _mean_	K_Q mean_
		K	kN	kN	N/N	MPa·m^0,5^
AA2519	CT	293	7.77	9.34	1.21	40.32
		77	8.14	9.66	1.19	40.36
Ti6Al4V	CT	293	15.23	17.46	1.14	74.58
		77	14.70	14.86	1.01	70.73
AA2519–AA1050–Ti6Al4V	CT	293	8.98	9.88	1.10	47.08
		77	7.78	8.11	1.04	42.11
	SENB	293	6.93	8.45	1.24	49.13
		223	6.30	7.62	1.21	44.70

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
