# Peer review of "Investigation of the Fracture Process of Explosively Welded AA2519–AA1050–Ti6Al4V Layered Material"

_materials, 2020, doi:10.3390/ma13102226_

Round 1
Reviewer 1 Report
The purpose of the present papers was to analyze the cracking process of Al-Ti laminated material with excellent mechanical property. The general idea of the article is a novelty and will be a very interesting topic for the audience of this journal. This AA2519-AA1050-Ti6Al4V layered material can be used as raw material in constructing ballistic protection shields for light military vehicles and not only. These results thereby provide an advance towards the current knowledge. The abstract is self-explanatory and totally reflects the content of the article. The Materials and Methods section is very well organized, the discussion section clearly states this discovery and therefore points out the added value of this research. Conclusions are well presented and describe the advances that were made as an outcome of the present paper, together with the perspectives for future research in the field.

Author Response
Response to Reviewer 1 Comments:
Thank you for the positive evaluation of our manuscript.
Reviewer 2 Report
as from my previous message
Author Response
Response to Reviewer 2 Comments:
Thank you for the positive evaluation of our manuscript
Reviewer 3 Report
Dear authors, thank you for submiting your interesting paper. After review, I have some remarks. Please, correct it.
Line 34 – I recommend using the term metallurgical properties instead of metallurgy properties.
Line 58 – I recommend mentioning the full name of the abbreviation CT and in the case of CTOD-R as well (Line 67).
Line 82 – Please, write aluminum-lithium alloys instead of aluminum and lithium alloys.
Line 131 – Please, check the Table 4 because in manuscript is divided by two pages.
Line 145 – Please, specify the welding parameters, used explosive, explosive thickness, detonation velocity and stand-off distance. What material was stable, and which one was the flyer plate.
Line 198 – Legend in the figure 3 is unclear, it is not obvious which temperature corresponds to red and blue line.
Line 269 – What does the abbreviation SENB mean?
Line 294 – Please, add scales to the figures 7a – 7d.
Line 311 – Fig. 8, please add the name of y axis on the linescan.
Line 331 – Fig. 9, please designate the materials in the figure.
Author Response
Response to Reviewer 3 Comments:
- Comment No 1:
Line 34 – I recommend using the term metallurgical properties instead of metallurgy properties.
Response to comment No 1:
Term ‘metallurgy’ has been changed to ‘metallurgical’ according to Reviewer’s remark (Line 37 in tracked version of manuscript).
- Comment No 2:
Line 58 – I recommend mentioning the full name of the abbreviation CT and in the case of CTOD-R as well (Line 67).
Response to comment No 2:
Full name of CT specimen was introduced to manuscript. ‘CT’ was replaced by ‘compact tension (CT) specimen’ (Line 76).
Similarly, ‘CTOD-R curve’ has been replaced by ‘crack tip opening displacement resistance curve (CTOD-R)’ (Line 85).
- Comment No 3:
Line 82 – Please, write aluminum-lithium alloys instead of aluminum and lithium alloys.
Response to comment No 3:
The incorrect wording 'aluminum and lithium' has been replaced by the correct description 'aluminum-lithium' (Line 100).
- Comment No 4:
Line 131 – Please, check the Table 4 because in manuscript is divided by two pages.
Response to comment No 4:
Dividing of the Table 4 was probably caused by different Word editor. In our version Table 4 is located in single page.
- Comment No 5:
Line 145 – Please, specify the welding parameters, used explosive, explosive thickness, detonation velocity and stand-off distance. What material was stable, and which one was the flyer plate.
Response to comment No 5:
Description of welding parameters was incorporated into manuscript (Lines 161-170):
'The explosive Saletrol (based on ammonium sulfate and hydrocarbon fuel) was used in the welding process. Testing plates were produced using the explosive material at a detonation velocity in the range of 1850–2000 m/s, at variable bonding parameters falling within the range of 420–620 m/s (for plates collision speed) and a collision angle of approximately 15°. The parallel plating configuration was applied, where the base layer was a 5 mm thick Ti6Al4V alloy sheet and the overlaid layer (flayer) was a 5 mm thick AA2519 alloy sheet with an approximately 0.2 mm thick unilaterally rolled soft layer of AA1050 aluminum alloy. The distance between the welded layers was 5 mm.'
- Comment No 6:
Line 198 – Legend in the figure 3 is unclear, it is not obvious which temperature corresponds to red and blue line.
Response to comment No 6:
Indeed legend in Figure 3b had an error and has been corrected. Additional legend was added to Figure 3d to improve the readability of the chart.
- Comment No 7:
Line 269 – What does the abbreviation SENB mean?
Response to comment No 7:
SENB is a name of the Single Edge Notched Bend specimen. Full name of SENB specimen was added at first occurrence of this type of specimen in the manuscript (line 189).
- Comment No 8:
Line 294 – Please, add scales to the figures 7a – 7d.
Response to comment No 8:
Scales have been added to all images showed in Figure 7.
- Comment No 9:
Line 311 – Fig. 8, please add the name of y axis on the linescan.
Response to comment No 9:
Name of axis y has been added to Figure 8. It is ‘cps’ (counts per second).
- Comment No 10:
Line 331 – Fig. 9, please designate the materials in the figure.
Response to comment No 10:
The layers of Al/Ti laminate showed in Figure 9 have been signed.
Thank you for all remarks.
Reviewer 4 Report
The authors studied the cracking process of explosive welded layered AA2519-AA1050-Ti6Al4V (Al-Ti laminate) at room and low temperatures. They measured the fracture toughness and found that the delamination cracking is the primary mode of fracture in layered laminates. The results are convincing, therefore paper can be accepted after following minor changes:
1) The abstract need to slightly modified to highlight the most important conclusions while ignoring the less important method details
2) The author may discuss other mechanical properties and/or functional properties of structural layered materials in the introduction section. e.g. "A deformation-processed Al-matrix/Ca-nanofilamentary composite with low density, high strength, and high conductivity"
3) The processing method has a huge impact on the bonding strength of the layered composite. This will cause the different fracture mode of the layered composite. The author can briefly discuss this bonding strength effect as described in this paper "Latest developments in modeling and characterization of joining metal based hybrid materials"
Author Response
Response to Reviewer 4 Comments:
- Comment No 1:
The abstract need to slightly modified to highlight the most important conclusions while ignoring the less important method details.
Response to comment No 1:
‘Abstract’ was rearranged and supplemented according to Reviewer’s remark.
- Comment No 2:
The author may discuss other mechanical properties and/or functional properties of structural layered materials in the introduction section. e.g. "A deformation-processed Al-matrix/Ca-nanofilamentary composite with low density, high strength, and high conductivity"
Response to comment No 2:
The information on the possibility of obtaining special properties of layered materials such as corrosion resistance, thermal conductivity and electrical conductivity has been added to the 'Introduction' (Line 45).
- Comment No 3:
The processing method has a huge impact on the bonding strength of the layered composite. This will cause the different fracture mode of the layered composite. The author can briefly discuss this bonding strength effect as described in this paper "Latest developments in modeling and characterization of joining metal based hybrid materials"
Response to comment No 3:
Thank you to the Reviewer for pointing out interesting paper, in which a number of experimental and analytical models for several bonding types of metal-metal composites were presented and discussed.
According to Reviewer's suggestion, brief description of the possibility of modelling the interfacial bonding strength in layered materials was added to the manuscript (Lines 63).
Thank you for all remarks.
Reviewer 5 Report
The topic presented and discussed in the present manuscript is of interest for our society and also for scientific point of view. Congrats for the figure, they are at high quality!
I beleive, the manuscript can be published in the Journal, after the corrections of some minor typos encountered along it.
Author Response
Response to Reviewer 5 Comments:
Thank you for the positive evaluation of our manuscript. We have corrected typos encountered along it.